

# The antimicrobial action of polyaniline involves production of oxidative stress while functionalisation of polyaniline introduces additional mechanisms

Julia Robertson[1], Marija Gizdavic-Nikolaidis[2], Michel K. Nieuwoudt[2] and Simon Swift[1]

[1] Department of Molecular Medicine and Pathology, University of Auckland, Auckland, New Zealand
[2] School of Chemistry, University of Auckland, Auckland, New Zealand

## ABSTRACT

Polyaniline (PANI) and functionalised polyanilines (fPANI) are novel antimicrobial agents whose mechanism of action was investigated. *Escherichia coli* single gene deletion mutants revealed that the antimicrobial mechanism of PANI likely involves production of hydrogen peroxide while homopolymer poly(3-aminobenzoic acid), P3ABA, used as an example of a fPANI, disrupts metabolic and respiratory machinery, by targeting ATP synthase and causes acid stress. PANI was more active against *E. coli* in aerobic, compared to anaerobic, conditions, while this was apparent for P3ABA only in rich media. Greater activity in aerobic conditions suggests involvement of reactive oxygen species. P3ABA treatment causes an increase in intracellular free iron, which is linked to perturbation of metabolic enzymes and could promote reactive oxygen species production. Addition of exogenous catalase protected *E. coli* from PANI antimicrobial action; however, this was not apparent for P3ABA treated cells. The results presented suggest that PANI induces production of hydrogen peroxide, which can promote formation of hydroxyl radicals causing biomolecule damage and potentially cell death. P3ABA is thought to act as an uncoupler by targeting ATP synthase resulting in a futile cycle, which precipitates dysregulation of iron homeostasis, oxidative stress, acid stress, and potentially the fatal loss of proton motive force.

## INTRODUCTION

Conducting polymers (CPs) are a class of polymeric materials that have electronic and ionic conductivity (*Macdiarmid, 2002*; *Ravichandran et al., 2010*). Polyaniline (PANI) and its derivatives comprise an important and widely studied class of conducting polymers (*Macdiarmid, 2002*; *Dhand et al., 2011*). Polyanilines are deemed to be derived from a polymer consisting of varying relative amounts of reduced and/or oxidised aniline subunits (Fig. 1A) (*Macdiarmid, 2002*; *Stejskal, Sapurina & Trchová, 2010*). The emeraldine state ('half-oxidised') has the greatest stability at room temperature. Application of a dopant ion to emeraldine base (EB) imparts conductivity and the resulting electrically conductive

Corresponding author
Simon Swift, s.swift@auckland.ac.nz

**A**

reduced unit                                    oxidised unit

Legend :

$y = 1$          Leucoemeraldine base (LEB)

$y = 0.5$       Emeraldine base (EB)

$y = 0$          Pernigraniline base (PNB)

**Emeraldine salt (ES)**

**B**

**Figure 1 Structures of PANI and P3ABA.** PANI (A) consists of varying relative numbers of reduced and/or oxidised aniline subunits, which determine the average oxidation state (defined as $1 - y$). Reduced polymers ($y = 1$), called leucoemeraldine, have an average oxidation state of 0. 'Half-oxidised' polymers ($y = 0.5$), called emeraldine, have an overall oxidation state of 0.5. Oxidised polymers ($y = 0$), called pernigraniline, have an average oxidation state of 1. P3ABA (B) is a homopolymer of 3-aminobenzoic acid (3-ABA). ABAs consist of an aniline benzene ring substituted with a carboxylic acid group (−COOH).

doped form, emeraldine salt (ES), is considered the most applicable form of PANI (*Ravichandran et al., 2010*; *Dhand et al., 2011*).

The utilisation of PANI is restricted by its insolubility in common solvents (*Genies et al., 1990*; *Gizdavic-Nikolaidis et al., 2011*). Functionalised polyanilines (fPANI) are synthesised from functionalised aniline monomers, and exhibit improved solubility and processability. (*Su & Kuramoto, 2000*; *Gizdavic-Nikolaidis et al., 2010b*; *Gizdavic-Nikolaidis et al., 2010a*; *Ravichandran et al., 2010*). A promising fPANI candidate is poly(3-aminobenzoic acid),

P3ABA (Fig. 1B) (*Salavagione et al., 2004*; *Stejskal, Sapurina & Trchová, 2010*; *Gizdavic-Nikolaidis et al., 2011*).

PANI and fPANI are novel antimicrobial agents that are active against a broad range of bacteria including *Escherichia coli*, *Staphylococcus aureus*, *Pseudomonas aeruginosa*, *Enterococcus faecalis*, and *Campylobacter jejuni* (*Shi et al., 2006*; *Gizdavic-Nikolaidis et al., 2011*; *Prasad et al., 2012*; *Kucekova et al., 2013*; *Dhivya, Vandarkuzhali & Radha, 2015*). PANI and fPANI have potential to be incorporated into surfaces or used as surface coatings for infection control and food safety applications (*Robertson, Gizdavic-Nikolaidis & Swift, 2018*). The mechanistic basis of antimicrobial activity is hypothesised to involve electrical conductivity, which can mediate contact with the negatively charged bacterial cell surface through electrostatic adherence (*Seshadri & Bhat, 2005*; *Shi et al., 2006*; *Gizdavic-Nikolaidis et al., 2011*). Determination of the mode of action of an antimicrobial agent is of value (*Kohanski, Dwyer & Collins, 2010*; *Green, Fulghum & Nordhaus, 2011*). Knowledge of mechanism of action aids in understanding and interpreting the antimicrobial activity in experimental assays (*Green, Fulghum & Nordhaus, 2011*). Rational improvement of an antimicrobial agent, such as via functionalisation of PANI, is reliant on elucidation of antimicrobial mechanism (*Brazas & Hancock, 2005*; *Kohanski, Dwyer & Collins, 2010*; *Grass, Rensing & Solioz, 2011*). The possibility of the emergence and spread of resistant organisms may also be informed from the mode of action of an antimicrobial agent (*Brazas & Hancock, 2005*; *Grass, Rensing & Solioz, 2011*).

The antimicrobial mechanism of an fPANI, a homopolymer of sulfanilic acid (PSO$_3$H), has been previously examined. Initial hypotheses pertaining to fPANI action were generated following analysis of the results of a transcriptomic analysis of *E. coli* MG1655 sub-lethally challenged with PSO$_3$H (*Gizdavic-Nikolaidis et al., 2011*). The interpretation of the transcriptomic analysis assumes that for the challenged cells upregulated genes are involved in the responses to any stresses experienced; whereas genes downregulated are possible targets. The PSO$_3$H challenged *E. coli* demonstrated upregulation of stress response genes involved in defence against oxidative and periplasmic stress as well as genes associated with iron homeostasis (*Gizdavic-Nikolaidis et al., 2011*). Expression of some genes in the OxyR regulon (*Anjem, Varghese & Imlay, 2009*), which is activated by hydrogen peroxide (H$_2$O$_2$) stress, were increased, such as *trxC*, *grxA*, *mntH,* and *sufB*. The protein products of these genes act to restore redox balance (Trx2, Grx1) and reactivate or repair oxidatively damaged enzymes (MntH and SufB) (*Anjem, Varghese & Imlay, 2009*). A few genes in the SoxRS regulon (*Gu & Imlay, 2011*), which responds to superoxide-based stress, were also upregulated, such as *soxR*, *soxS*, and *fumC*. Periplasmic stress response genes included *spy*, *asr,* and *cpxP*, which encode proteins that act to reduce protein misfolding and aggregation (*Seputiene et al., 2003*; *Raivio, Leblanc & Price, 2013*). Upregulated genes involved in iron homeostasis (*Liu, Bauer & Imlay, 2011*) included *sufC* and *fes*. The *sufC* gene is a member of the SUF pathway and encodes an ATPase that facilitates transfer of nascent iron sulphur (FeS) clusters to apo-protein substrates or acts as a carrier protein during repair of oxidised FeS clusters (*Bandyopadhyay, Chandramouli & Johnson, 2008*; *Mettert & Kiley, 2015*). The

*fes* gene encodes an esterase enzyme that mediates release of iron from the enterobactin-iron complex during iron import (*Andrews, Robinson & Rodríguez-Quiñones, 2003*; *Adler et al., 2014*).

Downregulation of genes was also identified in the transcriptomic analysis (*Gizdavic-Nikolaidis et al., 2011*). These genes encoded proteins involved in central metabolism and energy generation including tricarboxylic acid (TCA) cycle genes, such as *sdhB* and *aceA* (*Martínez-Gómez et al., 2012*). The gene *sdhB* encodes the b subunit of succinate dehydrogenase (SDH) (*Cheng et al., 2006*; *Tran et al., 2006*). SDH oxidises succinate generating fumarate in the TCA cycle. Electrons from this oxidation are shuttled into the electron transport chain (ETC), providing a functional link between the TCA cycle and the ETC (*Cheng et al., 2006*; *Tran et al., 2006*). The SdhB subunit contains an $[4Fe-4S]^{2+}$ cluster, which is susceptible to oxidation by $H_2O_2$ resulting in dissociation of an iron atom and loss of enzyme activity (*Tran et al., 2006*; *Imlay, 2013*). Isocitrate lyase, encoded by the *aceA* gene, is a glyoxylate shunt enzyme that cleaves isocitrate to generate succinate and glyoxylate (*Blankenhorn, Phillips & Slonczewski, 1999*; *Kim & Copley, 2007*). Isocitrate lyase expression is induced by oxygen (*Blankenhorn, Phillips & Slonczewski, 1999*). The transcriptional response of *E. coli* MG1655 to the fPANI led to the development of several hypotheses pertaining to potential mechanisms of action including induction of oxidative stress, dysregulation of iron homeostasis, targeting of metabolic and respiratory enzymes, and disruption of membrane integrity.

Antimicrobial activity can be influenced by the physical properties of the polymer chain, including chain length and monomer type (*Gizdavic-Nikolaidis et al., 2012*). Therefore, the antimicrobial mechanisms of PANI or P3ABA may differ from that of $PSO_3H$ and required separate determination. In this study we characterise the antimicrobial mechanism of the more potent ES forms of PANI and P3ABA (*Shi et al., 2006*; *Gizdavic-Nikolaidis et al., 2011*) against *E. coli* 25922, a standard antibiotic sensitivity testing strain (*Wiegand, Hilpert & Hancock, 2008*). Previous work has demonstrated PANI and P3ABA activity against *E. coli* and *Staphylococcus aureus* in suspension and when incorporated into surfaces (*Robertson, Gizdavic-Nikolaidis & Swift, 2018*). We present evidence that supports an antimicrobial action for PANI that involves production of $H_2O_2$, which can promote formation of hydroxyl radicals causing biomolecule damage and potentially cell death. P3ABA was demonstrated to target ATP synthase, acting as an uncoupler, likely resulting in a futile cycle, which can promote dysregulation of iron homeostasis, oxidative stress, acid stress, and potentially the fatal loss of proton motive force.

## MATERIALS & METHODS

### Bacterial strains

The antimicrobial mechanism of PANI and P3ABA was initially investigated using *E. coli* single gene deletion mutants from the Keio collection (*Baba et al., 2006*; *Tamae et al., 2008*). The parent strain, *E. coli* K-12 strain BW25113 (F-Δ(*araD-araB*)567, Δ*lacZ* 4787(::rrnB-3), λ-, rph-1, Δ(*rhaD-rhaB*)568, *hsdR*514) and a selection of the isogenic *E. coli* deletion mutants were used in this work. These deletion mutants were *E. coli* JW3914-1 Δ*katG*,

*E. coli* JW1721-1 $\Delta katE$, *E. coli* JW0598-2 $\Delta ahpC$, *E. coli* JW3879-1 $\Delta sodA$, *E. coli* JW1648-1 $\Delta sodB$, *E. coli* JW1638-1 $\Delta sodC$, *E. coli* JW0669-2 $\Delta fur$, *E. coli* JW2566-1 $\Delta trxC$, *E. coli* JW0833-1 $\Delta grxA$, *E. coli* JW5195-1 $\Delta tonB$, *E. coli* JW2514-4 $\Delta iscS$, *E. coli* JW0714-1 $\Delta sdhB$, *E. coli* JW3715-1 $\Delta atpE$, *E. coli* JW1732-1 $\Delta spy$, *E. coli* JW5826-1 $\Delta asr$, *E. coli* JW2669-1 $\Delta recA$ and *E. coli* JW1625-1 $\Delta nth$.

*E.coli* ATCC 25922 (referred to as *E. coli* 25922) was selected for further characterisation of the action of PANI and P3ABA because it is a standard antibiotic sensitivity testing strain (*Wiegand, Hilpert & Hancock, 2008*). All strains were grown at 37 °C, with shaking at 200 rpm where appropriate.

## Media and chemicals

PANI and P3ABA were synthesised via chemical oxidation of aniline and 3-aminobenzoic acid monomers, respectively (*Gizdavic-Nikolaidis et al., 2011*). Cell biology reagents were purchased from Sigma-Aldrich. General reagents included sodium chloride, methylene blue, resazurin sodium salt, agar, L-cysteine, glycerol, potassium nitrate, sodium succinate dibasic hexahydrate and 30% w/v $H_2O_2$.

Bacteria were cultured in LB broth (BD) or in minimal media. Minimal A medium was used to support growth in a minimal environment providing only essential nutrients. A 5× minimal A solution was made according to the following: 5 g $(NH_4)_2SO_4$, 22.5 g $KH_2PO_4$, 52.5 g $K_2HPO_4$, 2.5 g sodium citrate.$2H_2O$. After autoclaving, this solution was diluted to 1× with sterile water and the following sterile solutions, per litre: 1 ml 1 M $MgSO_4.7H_2O$, 0.1 ml 0.5% thiamine plus the carbon source (10 ml of 70% glycerol solution or 10 ml of 40% succinate solution per litre). A 20% casamino acid solution was used at 0.1% in minimal A salts media where needed to overcome growth defects.

A 0.2 M deferoxamine mesylate (DF) solution was prepared by adding 131.36 g of DF per litre to water and adjusting the pH to 7.4 using KOH. A 20 mM Tris-HCl-10% glycerol (pH 7.4) buffer solution was prepared by adding 2.4 g Tris ultrapure and 100 ml glycerol per litre of water and adjusting the pH using concentrated HCl.

## Preparation of PANI and P3ABA suspensions

PANI was finely ground using a mortar and pestle. This insoluble powder requires shaking at 200 rpm to stay in suspension. Reflecting the improved solubility of P3ABA, this polymer was suspended in broth using the QSonica Q700 Sonicator at the following settings: amplitude 30, elapsed time 10 s, repeat 4×.

## Sensitivity of *E. coli* deletion mutants to PANI and P3ABA suspensions

Suspensions of PANI and P3ABA were prepared at 2% (w/v) for a final concentration of 1% in antimicrobial challenges. Turbid cultures of test bacteria (parent strain plus deletion mutant strains) were diluted to $10^6$ CFU/ml in LB broth or minimal A salts with 0.4% succinate. The *E. coli* deletion mutants missing genes associated with metabolism (*E. coli* $\Delta iscS$, *E. coli* $\Delta tonB$, and *E. coli* $\Delta sdhB$) displayed growth defects in minimal media, which were overcome by supplementing the minimal media with 0.1% casamino acids for both

**Table 1** **Summary of the relative sensitivities of *E. coli* deletion mutants to 1% PANI and 1% P3ABA suspensions in LB broth and minimal A salts with 0.4% succinate.** Only deletion mutants that were significantly more or less sensitive to PANI or P3ABA treatment compared to the parent strain are presented. Blank spaces in the table represent non-significant differences for these strains. *E. coli* deletion mutants that were not significantly different for either treatment were not included in the table. The data the table was generated from is included in the raw data file.

| Mutant strain | | LB broth | | Minimal media | |
|---|---|---|---|---|---|
| | | PANI | P3ABA | PANI | P3ABA |
| Oxidative stress response | Δ*katG* | More | | | |
| | Δ*ahpC* | Less | | | |
| | Δ*grxA* | | | | More |
| | Δ*tonB* | Less | | | More |
| Iron homeostasis | Δ*iscS* | More | | | More |
| | Δ*fur* | | | | More |
| Metabolism and respiration | Δ*atpE* | | Less | | |
| | Δ*sdhB* | | | | More |
| Periplasmic stress response | Δ*spy* | More | Less | | |
| | Δ*asr* | | | | More |
| DNA repair | Δ*recA* | Less | | | |

the mutant strain and the parent strain. *E. coli* Δ*atpE* was unable to grow in minimal media even with supplementation of casamino acids and therefore was only tested in rich media.

Samples of 500 µl of either PANI suspension, P3ABA suspension, and growth media (no antimicrobial control) were inoculated with 500 µl of diluted culture. The experimental samples were incubated at 37 °C with 200 rpm agitation. For challenge in rich media, PANI treated cells were rescued at 2 h and 4 h while P3ABA treated cells were rescued at 0.5 h and 1 h reflecting the difference in activity levels against target bacteria in LB broth. For challenge in minimal media, both PANI and P3ABA treated cells were recovered after 1 h of incubation. At these time points, an aliquot from each experimental sample was enumerated using the drop plate method on LB agar plates. Following incubation, colonies were counted and CFU/ml was calculated. At least three biological replicates were obtained.

Linear regression analysis was used to compare the sensitivity of the parent strain to that of each *E. coli* deletion mutant for both PANI and P3ABA. Data was graphed in a scatter plot generated with viable cell counts post-treatment (CFU/ml) represented on the *y*-axis and time (h) represented on the *x*-axis. Linear regression was used to fit a straight line (regression line) through the data for the categorical factor (bacterial gene mutation) generating the best-fit value of the slope and intercept. An analysis of covariance (ANCOVA) was used to compare the regression lines from the parent strain and a mutant strain to determine if the effect of bacterial gene mutation on viable cell count post-treatment represented a statistically significant increase or decrease in sensitivity. Significant differences in sensitivity between the parent strain and a mutant strain are represented in Table 1.

## Electron paramagnetic resonance (EPR) spectroscopy

The protocol used was based on the whole cell EPR assay to detect free iron in *E. coli* (*Woodmansee & Imlay, 2002*; *Liu & Imlay, 2013*). A turbid culture of *E. coli* 25922 was diluted 1:100 in 6 × 200 ml LB broth aliquots and grown aerobically at 37 °C until log phase (optical density at 600 nm, $OD_{600}$, of 0.4–0.6) was reached. All cultures were centrifuged at 3,000× g at 4 °C for 5 min. Two cultures were assigned to P3ABA treatment, to $H_2O_2$ treatment or left untreated as a control. A 5% P3ABA suspension in LB broth was prepared.

The cultures for P3ABA treatment were resuspended in 8 ml LB broth, 1 ml 5% P3ABA suspension and 1 ml of 0.2 M DF solution. Final concentrations of P3ABA and DF were 0.5% (w/v) and 0.02 M, respectively. The P3ABA treated cultures were incubated at 37 °C with shaking for 30 min. The cultures for $H_2O_2$ treatment were resuspended in 8 ml LB broth, 1 ml 200 mM $H_2O_2$, and 1 ml of 0.2 M DF solution, and incubated for 5 min. Untreated cultures were resuspended in 9 ml LB broth and 1 ml of 0.2 M DF solution, and incubated for 30 min.

All cultures were centrifuged at 3,400× g at 4 °C for 5 min twice. They were washed with 10 ml ice cold 20 mM Tris-HCl-10% glycerol pH 7.4 buffer and finally resuspended in 200 µl of buffer. A 200 µl aliquot of each cell suspension was added to a 4 mm quartz tube and frozen in a dry ice-ethanol bath. Samples were stored at −80 °C prior to EPR analysis. A 10 µl aliquot of each sample was diluted 1:100 in buffer and the $OD_{600}$ was determined.

The EPR standard was established by dissolving 0.15 g of anhydrous $Fe_2(SO_4)_3$ in 20 mM Tris-HCl-1 mM DF-10% glycerol (pH 7.4). The ferric sulfate solution was diluted to a range of concentrations (6 µM, 17 µM, 36 µM, 76 µM, and 375 µM). The absorption coefficient, $\varepsilon_{420} = 2.865$ mM$^{-1}$ cm$^{-1}$, was used precisely quantitate the $Fe^{3+}$ concentrations using the Beer-Lambert law. The ferric sulfate solutions were loaded into EPR tubes and handled in the same manner as the cultures.

The EPR signals were recorded using the JEOL JES-FA 200 EPR spectrometer. The recording temperature was set at −196 °C by liquid nitrogen. The measurement parameters were as follows: magnetic centre field 151.419 mT, microwave frequency 9,072.85 MHz, microwave power 6.0 mW, sweep width 50 mT, modulation width 0.35 mT, amplitude/gain 200, time constant 0.10 s and scan time two min. EPR spectra were confirmed in two replicate experiments. The spin number of each sample was calculated from the second integral of the recorded EPR spectrum.

Data generated from measuring free iron using EPR is presented as a spectrum in which the size of the peak can be used to quantify the concentration of $Fe^{3+}$ (*Cammack & Cooper, 1993*; *Woodmansee & Imlay, 2002*). The area under the peak corresponds to the signal intensity (*Woodmansee & Imlay, 2002*). A large peak corresponds to a high level of free iron and a small peak corresponds to a normal low level (*Woodmansee & Imlay, 2002*). The concentration of iron (µM) for each sample was determined from iron buffer standard curve (*Woodmansee & Imlay, 2002*). From the two cultures, the median free iron value for each treatment was calculated. At least three biological replicates were obtained. The Kruskal–Wallis test with Dunn's multiple comparison post-hoc test was used analyse difference between the free iron levels of P3ABA treated, $H_2O_2$ treated, and untreated cells.

## Assay to determine the activity of PANI and P3ABA suspensions in aerobic and anaerobic conditions against *E. coli*

*E. coli* 25922 was challenged with PANI or P3ABA suspensions in aerobic and anaerobic conditions to determine the minimum inhibitory concentration (MIC) and minimum bactericidal concentration (MBC) (*Andrews, 2001*; *Dwyer et al., 2014*). GasPak$^{TM}$ EZ Small Incubation Containers were used in conjunction with GasPak$^{TM}$ EZ Anaerobic Container System sachets to create an anaerobic environment for antimicrobial testing. To prepare the growth media for antimicrobial assays in anaerobic conditions, a 2% cysteine stock solution was added to liquid growth media to a final concentration of 0.05% to create a reducing environment with less reactive oxygen species (ROS) present (*Park & Imlay, 2003*). The anaerobic environments used in this work were monitored using the redox indicator solutions, 0.0002% methylene blue and 0.0002% resazurin. These dyes appear blue when oxidised (indicating oxygen is present) and colourless when reduced (indicating oxygen is absent) (*Karakashev, Galabova & Simeonov, 2003*).

A range of concentrations of PANI and P3ABA in suspension were established. PANI is an insoluble powder that is not suitable for dilution from a stock solution. Each PANI suspension was set up separately by weighing the powder into 5 ml tubes and adding 500 µl of broth. A P3ABA stock suspension was established by adding 0.8 g of P3ABA to 5 ml of broth. The P3ABA stock suspension was diluted to 0.125% P3ABA using doubling dilution series. The suspensions were established at 2× the final desired concentration (which was achieved following inoculation with an equal volume). Final concentrations tested were 0.00391%–16% for PANI and 0.015625%–8% for P3ABA. 500 µl of each established suspension was aliquoted into a 5 ml tube and 500 µl of growth media was aliquoted to set up an untreated control. LB broth was used as a rich media while minimal A salts was used as a minimal media with 0.7% glycerol or 0.4% succinate as the carbon source. To support the growth of cells in anaerobic conditions, potassium nitrate was added at 20 mM to the minimal media. Nitrate can act as a terminal electron acceptor in *E. coli* facilitating anaerobic respiration (*Dwyer et al., 2014*). For treatment in aerobic conditions, no nitrate was present in the growth media.

An *E. coli* 25922 culture, suspensions for anaerobic challenge, media used for preparing the inoculum, and redox indicator tubes were incubated anaerobically at 37 °C with 200 rpm shaking. Aerobic suspensions and redox indicator tubes were incubated aerobically at 37 °C with 200 rpm shaking. Following a 24 h (for LB broth) or 48 h (for minimal media) incubation, a turbid anaerobic culture of test bacteria was diluted to $10^6$ CFU/ml in broth. The inoculum was retrospectively enumerated using the spread plate method.

Each aerobic and anaerobic PANI or P3ABA suspension (and the untreated control) was inoculated with 500 µl of diluted culture. The experimental samples were incubated aerobically and anaerobically (as appropriate). The MIC was defined as the lowest concentration of PANI or P3ABA that was able to inhibit the visible growth of test bacteria following a 24 h or 48 h treatment.

Tubes that were observed by eye to have no visible growth were selected for MBC testing. For this, 20 µl of the experimental sample was spread onto 6 LB agar plates. The spread plates were incubated at 37 °C for 16 h and growth on these plates was determined. When

countable colonies were present, the CFU/ml of the sample was calculated. Bactericidal activity was defined as 99.9% reduction in cell number relative to the starting inoculum. The MBC was defined as the lowest concentration of PANI or P3ABA that prevented the growth of test bacteria following subculture on LB agar plates. At least three biological replicates were obtained for each experiment.

The Mann–Whitney test was used to analyse the difference between the aerobic and anaerobic MIC, and the aerobic and anaerobic MBC of PANI and P3ABA against *E. coli* 25922. A *P* value of less than 0.05 is taken to signify that the differences between the two groups is true while a *P* value of more than 0.05 is taken to indicate that the distribution of data of both groups is the same. The Mann–Whitney test was also used to compare the difference between the influence of the presence of oxygen on the activity of PANI and P3ABA against *E. coli* 25922.

## Catalase protection assay

Suspensions of PANI and P3ABA in LB broth were prepared to achieve final concentrations of 2% and 0.5% (w/v), respectively. A $H_2O_2$ solution was prepared for a final concentration of 10 mM in 0.85% saline. A 500 U/ml solution of catalase was made in LB broth. A turbid culture of *E. coli* 25922 was diluted to $10^6$ CFU/ml in LB broth. 500 μl aliquots of diluted culture were challenged with PANI suspension, P3ABA suspension, $H_2O_2$ solution, or LB broth (untreated control) in the presence or absence of exogenous catalase. Experimental samples were incubated at 37 °C with 200 rpm agitation. PANI treated and P3ABA treated cells were enumerated at 2 h, and 0.5 h time points, respectively. $H_2O_2$ treated and untreated cells were enumerated at all time points using the drop plate method. At least three biological replicates were obtained. A repeated measures ANOVA was used to analyse the difference in cell viability of PANI treated, P3ABA treated, and $H_2O_2$ treated cells in the presence and absence of exogenous catalase.

## Statistical analysis

Statistical analysis for all experiments was performed using GraphPad Prism software version 6 (GraphPad Software, Inc., San Diego, CA, USA). A *P* value less than 0.05 was considered to represent statistical significance.

## Sensitivity of *E. coli* deletion mutants to PANI and P3ABA suspensions

Linear regression analysis was used to compare the sensitivity of the parent strain to that of each *E. coli* deletion mutant for both PANI and P3ABA. Linear regression was used to fit a regression line through the data generating the best-fit value of the slope and intercept. An analysis of covariance (ANCOVA) was used to compare the regression lines from the parent strain and a mutant strain to determine if the effect of bacterial gene mutation on viable cell count post-treatment represented a statistically significant increase or decrease in sensitivity.

## EPR spectroscopy

The Kruskal–Wallis test with Dunn's multiple comparison post-hoc test was used analyse difference between the free iron levels of P3ABA treated, $H_2O_2$ treated, and untreated cells.

### Activity of PANI and P3ABA suspensions in aerobic and anaerobic conditions

The Mann–Whitney test was used to analyse the difference between the aerobic and anaerobic MIC, and the aerobic and anaerobic MBC of PANI and P3ABA against *E. coli* 25922. The Mann–Whitney test was also used to compare the difference between the influence of the presence of oxygen on the activity of PANI and P3ABA against *E. coli* 25922.

### Catalase protection assay

A repeated measures ANOVA was used to analyse the difference in cell viability of PANI treated, P3ABA treated, and $H_2O_2$ treated cells in the presence and absence of exogenous catalase.

## RESULTS AND DISCUSSION

### Sensitivity of *E. coli* deletion mutants to PANI and P3ABA suspensions

Based on hypotheses generated from the transcriptomic analysis of *E. coli* cells challenged with an fPANI, the antimicrobial mechanism of PANI and P3ABA was investigated using selected deletion mutants from the Keio collection. Determination of the sensitivity of deletion mutants can be used to generate initial hypotheses pertaining to antimicrobial mechanism (*Liu et al., 2010*) or used to investigate existing mechanism hypotheses in a more targeted fashion (*Feld, Knudsen & Gram, 2012*).

The mutants were selected based on the results from a transcriptomic analysis performed on cells challenged with an fPANI (*Gizdavic-Nikolaidis et al., 2011*). We hypothesised that *E. coli* mutants lacking oxidative stress response genes (*E. coli* Δ*katG*, *E. coli* Δ*katE*, *E. coli* Δ*ahpC*, *E. coli* Δ*sodA*, *E. coli* Δ*sodB*, *E. coli* Δ*sodC*), genes involved in iron homeostasis (*E. coli* Δ*fur*, *E. coli* Δ*tonB*, *E. coli* Δ*iscS*), genes involved in periplasmic stress responses (*E. coli* Δ*spy*, *E. coli* Δ*asr*), and genes involved in DNA repair (*E. coli* Δ*recA*, *E. coli* Δ*nth*) would be more sensitive to PANI and P3ABA treatment. We also hypothesised that *E. coli* mutants lacking genes involved in metabolism and respiration (*E. coli* Δ*trxC*, *E. coli* Δ*grxA*, *E. coli* Δ*sdhB*, *E. coli* Δ*atpE*) would be less sensitive to PANI and P3ABA treatment.

The parent strain (*E. coli* K12 BW25113) and mutants were challenged with 1% PANI or 1% P3ABA in suspension, and enumerated at selected time points (*Tamae et al., 2008*; *Feld, Knudsen & Gram, 2012*). The sensitivities of the deletion mutants relative to the parent strain were determined (Table 1; Data S1) to infer mode of action and generate hypotheses to be tested more rigorously. Details on statistical analysis are outlined in the final paragraph in the 'Sensitivity of *E. coli* deletion mutants to PANI and P3ABA suspensions' section and in the 'Statistical analysis' section in the materials and methods section. Bacteria with mutations in genes involved in the defence against antimicrobial action will have increased sensitivity to treatment while mutation of genes encoding antimicrobial targets will exhibit a decreased sensitivity to treatment (*Brazas & Hancock, 2005*; *Tamae et al., 2008*; *Liu et al., 2010*).

The *E. coli* deletion mutants were first tested in LB broth, a complex rich medium (*Zhang & Greasham, 1999*; *Sezonov, Joseleau-Petit & D'Ari, 2007*). A subset of mutants was selected for testing in a defined minimal medium with 0.4 % succinate as the carbon source to focus on antimicrobial effects in cells deriving energy almost exclusively from oxidative phosphorylation (*Cheng et al., 2006*; *Hards et al., 2015*). The deletion mutants were selected for testing in minimal media based on the hypotheses generated from the testing in rich media.

A role for oxidative stress from the production of $H_2O_2$ in the antibacterial mode of action of PANI is supported by (a) the supersensitivity of *E. coli* $\Delta katG$, which is unable to scavenge endogenous $H_2O_2$ and respond to the oxidative stress, and (b) the decreased sensitivity of *E. coli* $\Delta ahpC$, which is missing the $H_2O_2$ scavenger utilised during normal growth leading to OxyR activation before application of the PANI stress (*Seaver & Imlay, 2001*; *Mishra & Imlay, 2012*; *Zhang, Alfano & Becker, 2015*). Free iron can propagate $H_2O_2$ stress by participating in Fenton reaction producing hydroxyl radicals (*Imlay, 2008*; *Chiang & Schellhorn, 2012*). The reduced sensitivity of *E. coli* $\Delta tonB$, which has disrupted iron import, infers induction of an oxidative stress state in PANI treated cells (*Touati et al., 1995*). Further evidence towards a $H_2O_2$ stress based mechanism of action for PANI is derived from the greater sensitivity of the *E. coli* $\Delta iscS$ mutant, which is unable to modify DNA for protection against oxidative stress (*An et al., 2012*). The supersensitivity of the Spy mutant suggests that at least some of the damage is occurring in the periplasm (*Raffa & Raivio, 2002*; *Quan et al., 2011*). There was no real effect of changing the medium. Therefore, the PANI antimicrobial mechanism is hypothesised to involve $H_2O_2$ generation linked to the dysregulation of iron homeostasis, which eventuates in damage to cellular biomolecules including periplasmic proteins.

The relative sensitivities of the selected *E. coli* deletion mutants to P3ABA treatment (Table 1; Data S1) suggested that P3ABA and PANI exert bactericidal activity through different mechanisms. Deletion of genes encoding $H_2O_2$ and superoxide scavenging enzymes, and proteins involved iron homeostasis did not change susceptibility to P3ABA treatment. P3ABA antimicrobial action appears to involve targeting of respiratory machinery, as reflected by the reduced sensitivity of *E. coli* $\Delta atpE$ to P3ABA treatment, which suggested that ATP synthase is targeted by P3ABA. Damage to ATP synthase would cause uncoupling of ATP synthesis from electron transport, which can result in futile cycling, and increased respiration (*Noda et al., 2006*; *Hards et al., 2015*). There was evidence to imply damaged proteins may accumulate in the periplasm following P3ABA exposure due to its proximity to the ETC and were inferred to happen based on the insensitivity of the Spy mutant (which was hypothesised to have upregulation of an additional extracytoplasmic stress response resulting in pre-adaption to P3ABA action) (*Raivio & Silhavy, 2001*).

Statistically significant differences between the sensitivity of *E. coli* deletion mutants and the parent strain to P3ABA treatment were only detected in minimal media (Table 1). Increases in sensitivity were attributed to *E. coli* $\Delta grxA$, *E. coli* $\Delta tonB$, *E. coli* $\Delta iscS$, *E. coli* $\Delta fur$, *E. coli* $\Delta sdhB$, and *E. coli* $\Delta asr$ treated with P3ABA suspension (Table 1) suggesting

that perturbation of metabolism and iron homeostasis, oxidative stress, and acid stress is occurring in P3ABA treated cells.

The supersensitivity of the Grx1 mutant to P3ABA infers that this protein is involved in defence against P3ABA action. One of the functions of the glutaredoxin system, of which Grx1 is a member, is to scavenge hydroxyl radicals, a highly reactive ROS that causes biomolecule damage (*Floyd & Lewis, 1983*; *Starke, Chock & Mieyal, 2003*). Therefore, the increased sensitivity of *E. coli* Δ*grxA* may reflect an inability to efficiently eliminate hydroxyl radicals produced during P3ABA antimicrobial action.

The increased sensitivity of *E. coli* Δ*fur* may be due to the role Fur plays in survival in iron limiting conditions, such as in minimal media (*Seo et al., 2014*). The effect of the absence of Fur may be exacerbated in minimal media conditions compared to in rich media sensitising the cells to the stress inducing actions of P3ABA. The increased sensitivity of *E. coli* Δ*tonB,* strain lacking a gene involved in the uptake of iron (*Moeck & Coulton, 1998*), supports the assertion that P3ABA disrupts iron homeostasis.

The increased sensitivities of *E. coli* Δ*sdhB* and *E. coli* Δ*iscS* might reflect P3ABA targeting of metabolism and respiration. *E. coli* Δ*sdhB* is lacking a gene encoding a subunit of SDH, which couples the metabolic TCA pathway to respiration in the ETC (*Tran et al., 2006*). The increased sensitivity of *E. coli* Δ*sdhB* to P3ABA suggests that loss of function of this important enzyme undermines the ability of the bacteria to survive the bactericidal challenge. *E. coli* Δ*iscS* is missing a gene from the ISC pathway that functions in *de novo* synthesis of FeS clusters for important metabolic and respiratory enzymes (*Schwartz et al., 2000*; *Djaman, Outten & Imlay, 2004*). The resulting loss of function of the ISC pathway is only partially compensated by the SUF pathway resulting in synthesis of only essential FeS clusters (*Mettert & Kiley, 2015*). The associated metabolic defects observed in *E. coli* Δ*iscS* may sensitise the cells to P3ABA-mediated damage of metabolic and respiratory machinery (*Schwartz et al., 2000*; *Djaman, Outten & Imlay, 2004*).

*E. coli* Δ*asr* is missing a gene encoding Asr, a small basic periplasmic protein that promotes survival under acid conditions (external pH of 4.5 or less) by putatively sequestering protons (*Seputiene et al., 2003*). The supersensitive phenotype suggests that in these conditions Asr confers resistance to the bactericidal action of P3ABA. The increased susceptibilities of mutants missing genes involved in metabolism and iron homeostasis to P3ABA in minimal media supports targeting of metabolic and respiratory machinery, which would be expected to elicit greater responses for cells that have the increased metabolic demand of growth on minimal media. The different responses seen in *E. coli* deletion mutants in rich and minimal media highlight the advantages for antimicrobial testing in more than one type of media. Overall, the antimicrobial susceptibility profiles for the *E. coli* deletion mutants in rich and minimal media led to the refinement of hypotheses pertaining to the antimicrobial mechanism of PANI and P3ABA.

## P3ABA treatment causes an increase in free iron in *E. coli*

Based on the results and interpretation of the initial mechanism studies, it was hypothesised that PANI and P3ABA treatment would cause an increase in intracellular free iron in *E. coli* cells. An increase in free iron could be indicative of damage to iron-sulfur clusters of

metabolic and respiratory enzymes as well as damage to mononuclear iron enzymes, possibly mediated through ROS (*Jang & Imlay, 2007*; *Anjem & Imlay, 2012*; *Liu & Imlay, 2013*; *Gu & Imlay, 2013*). The effect of P3ABA on free iron levels in *E. coli* cells was examined using electron paramagnetic resonance (EPR) spectroscopy (*Cammack & Cooper, 1993*; *Woodmansee & Imlay, 2002*; *Liu & Imlay, 2013*). EPR spectroscopy detects unpaired electrons, such as in ferric iron, permitting quantitation of free ferric iron levels (*Cammack & Cooper, 1993*; *Woodmansee & Imlay, 2002*). Since intracellular free iron can exist as $Fe^{2+}$ or $Fe^{3+}$, DF was added to treated cells as this iron chelator converts all free $Fe^{2+}$ to the $Fe^{3+}$ form permitting detection of the whole free iron pool (*Liu, Bauer & Imlay, 2011*). The influence of PANI on free iron levels in *E. coli* could not be determined due to the insolubility of PANI.

$H_2O_2$ was used as a positive control in this experiment as it is known to cause an increase in unincorporated iron (*Jang & Imlay, 2007*; *Anjem & Imlay, 2012*). Following from this, antimicrobial activity involving oxidative stress should be signified by an increase in free iron levels in treated cells. Therefore, to determine if induction of oxidative stress occurs as part of P3ABA antimicrobial action, free iron levels were measured in *E. coli* cells sublethally treated with 0.5% P3ABA (*Varghese et al., 2007*; *Liu & Imlay, 2013*; *Gu & Imlay, 2013*).

A large peak representing intracellular $Fe^{3+}$ was detected from $H_2O_2$ treated and P3ABA cells, which were greater than that of untreated cells. This is indicative of oxidative damage to susceptible iron containing enzymes and the resulting release of iron. The free iron concentrations in $H_2O_2$ treated, P3ABA treated, and untreated cells were determined from the spectra using the standard curve and are presented in Fig. 2. The median free iron concentration in $H_2O_2$ treated cells (90 $\mu M$) and P3ABA treated cells (146 $\mu M$) were larger than the concentration in untreated cells (22 $\mu M$). The difference between P3ABA treated cells and untreated cells was statistically significant (Kruskal–Wallis test, *P* value: less than 0.05, Dunn's multiple comparison test). These results suggest that P3ABA induces oxidative stress and the associated increase in unincorporated free iron in *E. coli* cells.

*E. coli* Δ*fur* and *E. coli* Δ*iscS* are missing pleiotropic genes that are involved not only in iron homeostasis but also in other cellular functions including control of biofilm formation for the former and sulfur trafficking for the latter (*Ezraty et al., 2013*; *Seo et al., 2014*). Therefore, a lack of increased sensitivity of these mutants to P3ABA could be explained by changes to cellular functions separate from control of iron homeostasis, which may influence susceptibility to antimicrobial treatment (*Jang & Imlay, 2007*). *E. coli* Δ*tonB* did not have altered sensitivity to P3ABA relative to the parent strain, which suggests that the major iron import system does not play a significant role in P3ABA mediated cell death but rather the increase in free iron may be attributed to disruption of internal iron sources. Perturbation of internal iron management would mean that prevention of iron import would not be protective against oxidative stress. A similar situation was found for gyrase inhibitor mediated cell death (*Dwyer et al., 2007*).
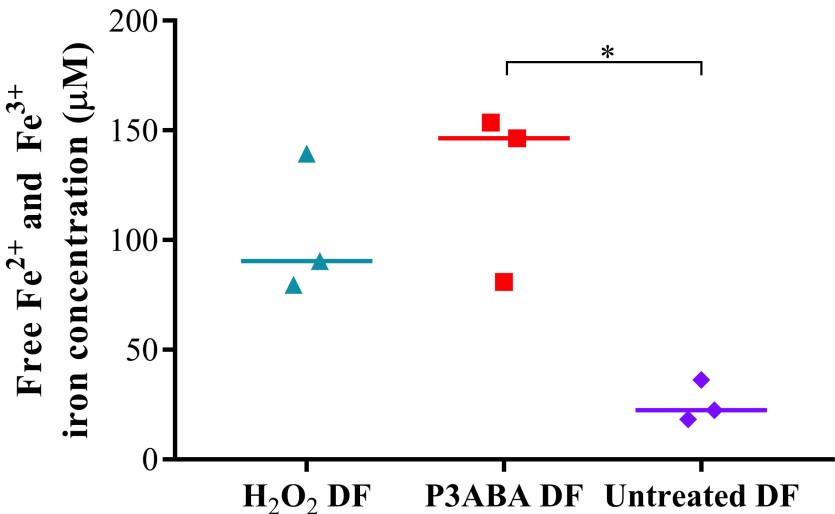

**Figure 2 Free iron concentration of $H_2O_2$ treated, P3ABA treated, and untreated *E. coli* samples.** Normalised EPR data showing the different free iron concentrations of 20 mM $H_2O_2$ treated, 0.5% P3ABA treated, and untreated *E. coli* cell samples. The treatment times for P3ABA and $H_2O_2$ were 30 min and 5 min, respectively. Each data point represents a biological replicate with a line plotted at the median. Statistical significance is represented by * (Kruskal-Wallis test, *P* value: less than 0.05, Dunn's multiple comparison post-hoc test).

## Activity of PANI and P3ABA suspensions in aerobic and anaerobic conditions

The relative sensitivities of *E. coli* deletion mutants to PANI in LB broth (Table 1) support the hypothesis that the antimicrobial mechanism of PANI involves induction of $H_2O_2$ based oxidative stress and the perturbation of iron homeostasis. In contrast, the relative sensitivities of *E. coli* deletion mutants to P3ABA (Table 1) suggest targeting of metabolic and respiratory machinery, specifically ATP synthase, and induction of acid stress. The increase in unincorporated iron in P3ABA treated cells (Fig. 2) supports disruption of internal iron homeostasis, possibly through oxidative stress. Perturbation of respiratory machinery function may result in production of ROS as a downstream effect (*Hards et al., 2015*). This is supported by the localisation of electrons and oxygen to the ETC, and the identification of the sources of ROS in *E. coli* (*Messner & Imlay, 1999*; *Korshunov & Imlay, 2006*; *Korshunov & Imlay, 2010*). Therefore, the activity of P3ABA may include production of ROS but at a lower extent than for PANI. The potential involvement of oxygen and its associated reactive species in the antimicrobial action of PANI and P3ABA warranted investigation.

To determine whether anaerobiosis was protective to the antimicrobial action of either PANI or P3ABA, the sensitivity of *E. coli* to these compounds in aerobic and anaerobic conditions was determined (*Keren et al., 2013*; *Dwyer et al., 2014*) as the MIC and MBC (*Wiegand, Hilpert & Hancock, 2008*; *Li et al., 2013*). The media in which bacterial cells are grown can have important effects on the growth and physiological state of the cells and therefore may influence sensitivity to inimical processes (*Kram*

& *Finkel, 2015*). *E. coli* cells growing in LB broth can generate ATP through substrate level phosphorylation, oxidative phosphorylation, or fermentation (*Barton, 2005*; *Alteri, Smith & Mobley, 2009*; *Farhana et al., 2010*). The increased growth rate and respiration of cells growing in LB broth means that they should be able to quickly respond to an external stress by synthesising defence proteins as the building blocks required to do this are found in the media (*Tao et al., 1999*). Minimal A is a defined medium that contains only the essential nutrients for growth (*Zhang & Greasham, 1999*). Glycerol and succinate can function as the sole carbon and energy source for *E. coli* cells growing in minimal media (*Tao et al., 1999*). All building blocks needed for *E. coli* to grow, such as amino acids, must be synthesised from the carbon source (*Tao et al., 1999*). Energy can only be derived from these non-fermentable carbon sources mostly through oxidative phosphorylation; however, substrate level phosphorylation is relevant for glycerol as well (*Jensen, Westerhoff & Michelsen, 1993*; *Boogerd et al., 1998*; *Bekker et al., 2009*).

### In aerobic conditions PANI is more active against *E. coli* 25922 in rich media and minimal media compared to anaerobic conditions

In rich media, PANI demonstrated greater activity in aerobic conditions compared to anaerobic conditions (Fig. 3A). The aerobic MIC and MBC (0.25% PANI) were less than the anaerobic MIC (3% PANI) and anaerobic MBC (8% PANI). A similar trend was seen for the activity against *E. coli* 25922 in minimal A salts media with 0.7% glycerol and minimal A salts media with 0.4% succinate as a carbon source (Figs. 3B–3C). In minimal media with glycerol, the aerobic MIC and MBC (0.015625% PANI) were lower than the anaerobic MIC (2% PANI) and MBC (8% PANI). In minimal media with succinate, the aerobic MIC (0.015625% PANI) and MBC (0.0625% PANI) were less than the anaerobic MIC (0.5% PANI) and MBC (8% PANI). For both minimal media types, the difference between the aerobic and anaerobic MIC and MBC of PANI against *E. coli* 25922 was statistically significant (Mann–Whitney test, $P$ value: less than 0.05). The reduced sensitivity of *E. coli* 25922 to PANI treatment in anaerobic conditions infers that oxygen and its associated reactive species are involved in the mode of action of PANI.

Aerobically, PANI had increased antimicrobial efficacy in minimal media compared to rich media as demonstrated by the lower MIC and MBC values in the former (Figs. 3A–3C). In rich media, the aerobic MIC and MBC (0.25% PANI) was lower than in minimal media with glycerol (MIC and MBC 0.015625% PANI) and in minimal media with succinate (MIC 0.015625% PANI, MBC 0.0625% PANI). The increased sensitivity of *E. coli* 25922 in minimal media to the bactericidal activity of PANI was abolished in anaerobic conditions (MBC for all media, 8% PANI) (Figs. 3A–3C), which indicates that the *E. coli* 25922 cells were more susceptible to the antimicrobial action of PANI in both aerobic and low nutrient conditions. The increased susceptibility to PANI action in minimal media compared to LB broth may reflect the *E. coli* cells ability to respond efficiently to PANI induced stress in rich media as cells in minimal media must synthesise all building blocks, the enzymes for which are susceptible to oxidative damage.

The aerobic MBC of PANI against *E. coli* 25922 in minimal media with succinate as the carbon source (0.0625% PANI) was higher than that in minimal media with glycerol as

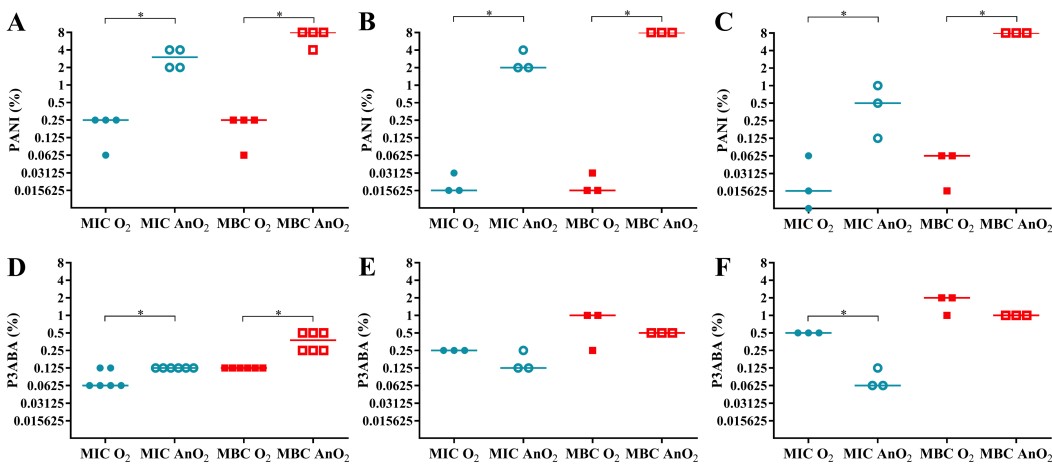

**Figure 3 The effect of oxygen on the activity of PANI and P3ABA against *E. coli* 25922.** The MIC (circles) and MBC (squares) of PANI and P3ABA in aerobic ($O_2$, filled data points) and anaerobic (An$O_2$, unfilled data points) conditions against *E. coli* 25922 are presented. The activity of PANI was determined in LB broth (A), minimal A salts with 0.7% glycerol (B), and minimal A salts with 0.4% succinate (C). The activity of P3ABA was determined in LB broth (D), minimal A salts with 0.7% glycerol (E), and minimal A salts with 0.4% succinate (F). Each data point represents a biological replicate with a line plotted at the median. Statistical significance is represented by * (Mann–Whitney test, *P* value: less than 0.05).

the carbon source (0.015625% PANI). The difference in susceptibility to PANI in these conditions may be reflective of the metabolic state of *E. coli* growing on the particular carbon source. The succinate concentration used to support cell growth in this experiment was ~15 mM. High concentrations of succinate (more than 8.8 mM) have been shown to reduce $H_2O_2$ formation by fumarate reductase, a flavin containing enzyme, by preventing autoxidation (*Messner & Imlay, 2002b*). The increased aerobic MBC of PANI against *E. coli* grown on succinate (compared to growth on glycerol) may be due to succinate somewhat reducing $H_2O_2$ production and following from this, susceptibility to PANI action. The lack of a difference in the bactericidal activity of PANI in the presence of succinate or glycerol in anaerobic conditions is supportive of this notion (Figs. 3A–3C).

## In aerobic conditions P3ABA is more active against *E. coli* 25922 in rich media and less active against *E. coli* 25922 in minimal media compared to anaerobic conditions

The hypothesis that *E. coli* cells would exhibit greater sensitivity to P3ABA in energy poor conditions (i.e., low nutrient and no oxygen conditions) compared to energy rich conditions was mostly not supported, with the exception of the low anaerobic MIC (0.0625%) in minimal media with succinate. It has been recently argued that ATP depletion following antimicrobial treatment does not cause bacterial cell death but rather delays the onset of killing (*Cook et al., 2014a*; *Lobritz et al., 2015*; *Hards et al., 2015*). It has been demonstrated that mycobacterial cells depleted of ATP do not loose viability (*Frampton et al., 2012*). The impact of ATP depletion on P3ABA action was indirectly examined by challenging *E. coli* cells grown on a non-glycolytic substrate, succinate, and *E. coli* cells grown on a glycolytic

substrate, glycerol (*Jensen et al., 1995*; *Sorgen, 1999*; *Hards et al., 2015*). The similar MIC and MBC values obtained for P3ABA with challenge on succinate and on glycerol carbon sources (Figs. 3E–3F) suggest that the ability to synthesise ATP by alternate methods does not prevent P3ABA bactericidal action. Consistent with this, treatment of *M. smegmatis* with a novel antitubercular drug that targets ATP synthase, bedaquiline, has been shown to result in downregulation of genes involved in glycolysis (*Hards et al., 2015*). Therefore, antimicrobial mechanisms of P3ABA separate to the putative ATP depletion may be involved in the bactericidal mode of action.

One important consideration for examining antimicrobial mechanism that involves targeting of metabolism and respiration is that the metabolic state of the cell may influence how it responds to bactericidal treatment (*Lobritz et al., 2015*). The bactericidal action of antimicrobial agents is associated with increased respiration while bacteriostatic action is characterised by suppressed cellular respiration (*Lobritz et al., 2015*). The bacteriostatic effect reduces ATP demand and is often the dominant effect blocking bactericidal action (*Lobritz et al., 2015*). Following from this, if cellular energy output is readily inhibited, such as in cells in energy poor conditions, antimicrobial action may result in inhibition of growth rather than bactericidal killing (*Lobritz et al., 2015*). Bacterial cells that are highly active may therefore be more susceptible to antimicrobial exposure because of accelerated respiration.

It has been postulated that bedaquiline exerts lethal activity by uncoupling of respiration-driven ATP synthesis causing increased respiration, loss of the proton gradient, and futile cycling (*Hards et al., 2015*). The MBC of P3ABA in LB broth (Fig. 3D; aerobic 0.125% P3ABA, anaerobic 0.375% P3ABA) was lower than that in minimal media with glycerol (Fig. 3E; aerobic 1% P3ABA, anaerobic 0.5% P3ABA) and in minimal media with succinate (Fig. 3F; aerobic 2% P3ABA, anaerobic 1% P3ABA). The greater activity of P3ABA in rich media relative to minimal media may be due to the lower energy state of the *E. coli* cells in minimal media. It is possible that the increased respiration and metabolic activity associated with *E. coli* cells in LB broth would predispose the cells to the bactericidal action of P3ABA (resulting in accelerated respiration and futile cycling) while the cells in minimal media may experience more inhibitory action (*Tao et al., 1999*). Indeed, the MIC of bedaquiline against *M. smegmatis* MC$^2$155 in LB broth (0.01 μg/ml) was lower compared to that in minimal media (0.025 μg/ml) (*Hards et al., 2015*). Succinate is a non-fermentable carbon source that necessitates ATP production to be carried out by the ETC (rather than by fermentation) compared to components in LB broth. The notable difference in the anaerobic MIC and MBC of P3ABA in minimal media with succinate (4 doubling dilutions) compared to that in other media (∼2 doubling dilutions) may be reflective of the effective inhibition of cells by P3ABA in a very energy-poor environment contributing to less effective bactericidal action (Figs. 3D–3F).

In this work, the hypothesis that the mode of action of P3ABA may eventuate in ROS production has been proposed. To investigate this hypothesis, *E. coli* 25922 was challenged with P3ABA suspensions in rich and minimal media in the presence and absence of oxygen. In rich media, the MIC and MBC of P3ABA against *E. coli* 25922 were higher in anaerobic conditions (MIC 0.125% P3ABA, MBC 0.375% P3ABA) compared to aerobic conditions

(MIC 0.0625% P3ABA, MBC 0.125% P3ABA) (Figs. 3D–3F). The greater MIC and MBC of P3ABA against *E. coli* 25922 in anaerobic conditions relative to aerobic conditions were statistically significant (Mann–Whitney test, *P* value: less than 0.05). The decreased activity of P3ABA in anaerobic conditions suggests that in rich media the antimicrobial activity of P3ABA involves a small amount of ROS production. Uncoupling activity, such as what is postulated to occur during P3ABA action, causes futile cycling, which is associated with increased oxygen consumption and production of ROS (*Adolfsen & Brynildsen, 2015*; *Hards et al., 2015*). Therefore, the minor role of induction of oxidative stress in P3ABA antimicrobial action may be due a downstream consequence of uncoupling electron transport from ATP synthesis. P3ABA activity was less affected by oxygenation compared to PANI activity, which implies that P3ABA acts through other more important antimicrobial mechanisms.

The trend observed in rich media was distinct from that observed in minimal media. Reduced activity of P3ABA was demonstrated when oxygen was present (Figs. 3E–3F). For minimal media with glycerol, the aerobic MIC (0.25% P3ABA) and MBC (1% P3ABA) were higher than the anaerobic MIC (0.125% P3ABA) and MBC (0.5% P3ABA). Similarly in minimal media with succinate, the aerobic MIC (0.5% P3ABA) and MBC (2% P3ABA) were greater than the anaerobic MIC (0.0625% P3ABA) and MBC (1% P3ABA). The difference between the MIC in minimal media with succinate in aerobic and anaerobic conditions was statistically significant (Mann–Whitney test, *P* value: less than 0.05) while all other differences observed for *E. coli* 25922 in minimal media were not statistically significant (Mann–Whitney test, *P* value: more than 0.05). These results suggest that induction of ROS-mediated stress is not occurring in *E. coli* cells treated with P3ABA in minimal media.

## PANI is less active in anaerobic conditions while P3ABA is less affected by the absence of oxygen

The overall aim of the determination of the MIC and MBC of PANI and P3ABA against *E. coli* in aerobic and anaerobic conditions was to examine the influence oxygen has on the activity of these antimicrobial agents. A higher MBC in anaerobic conditions compared to aerobic conditions is indicative of a reduction in bactericidal activity when no oxygen is present, which would suggest that, taken with other evidence, the antimicrobial mechanism involves induction of oxidative stress. The fold difference between the aerobic MBC and anaerobic MBC reflects the extent oxygenation influences lethal activity. The difference in activity between aerobic and anaerobic conditions is presented as a graph in Fig. 4 with the fold change in MBC between aerobic and anaerobic conditions on the *y*-axis and treatment on the *x*-axis. Greater activity in aerobic conditions (MBC $O_2$ less than MBC $AnO_2$) is represented by a number greater than 1 while greater activity in anaerobic conditions (MBC $O_2$ more than MBC $AnO_2$) is represented by a number less than 1. No difference in activity between aerobic and anaerobic conditions (MBC $O_2$ = MBC $AnO_2$) is represented by 1.

The activity of PANI against *E. coli* 25922 in all media tested was decreased in anaerobic conditions as indicated by a fold change of 32 and above (Fig. 4). In contrast to PANI, the

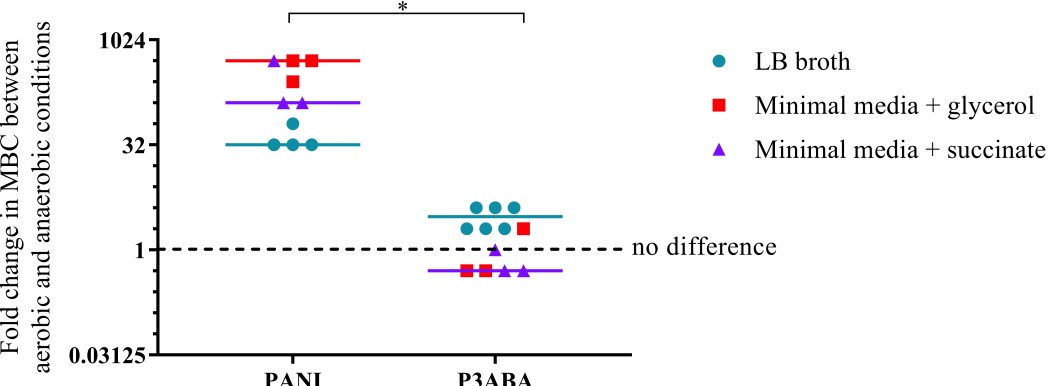

**Figure 4** **The difference between the MBC of PANI and P3ABA against *E. coli* 25922 in aerobic and anaerobic conditions.** The fold change in the MBC of PANI and P3ABA against *E. coli* 25922 in aerobic and anaerobic conditions. *E. coli* 25922 cells were challenged in LB broth (circles), minimal A salts with 0.7% glycerol (squares) and minimal A salts with 0.4% succinate (triangles). The dotted line ($y = 1$) represents no difference in the aerobic MBC and anaerobic MBC of PANI and P3ABA. Each data point represents a biological replicate with a line plotted at the median. Statistical significance is represented by * (Mann–Whitney test, *P* value: less than 0.05).

activity of P3ABA against these strains in LB broth was not affected as much by the presence and absence of oxygen (Fig. 4). The difference between the influence of the presence of oxygen on the activity of PANI compared to P3ABA against *E. coli* 25922 was statistically significant (Mann–Whitney test, *P* value: less than 0.05). These results demonstrate that the antimicrobial activity of PANI is more greatly influenced by the presence of oxygen compared to P3ABA, which supports the hypothesis that PANI mediates induction of oxidative stress.

## Effect of catalase on sensitivity of *E. coli* to PANI and P3ABA

The relative sensitivities of *E. coli* deletion mutants to PANI in LB broth (Table 1) support the hypothesis that the antimicrobial mechanism of PANI involves increased production of $H_2O_2$ and the perturbation of iron homeostasis, the result of which is endogenous oxidative stress. The reduced efficacy of PANI suspension against *E. coli* 25922 in anaerobic, compared to aerobic, conditions (Fig. 4) supported this hypothesis. P3ABA is hypothesised to target respiratory machinery, such as the ATP synthase, based on the greatly reduced sensitivity of *E. coli* Δ*atpE* and the loss of membrane potential measured in P3ABA treated *E. coli* cells (*Robertson, 2012*). There is limited evidence that P3ABA treatment causes oxidative stress in *E. coli* grown in a rich media (Figs. 3D–3F) but not in minimal media (Figs. 3E–3F). P3ABA antimicrobial action is less affected by the presence of oxygen compared to PANI (Fig. 4).

The role of oxidative stress in PANI and P3ABA action was further investigated by determining if exogenous catalase has a protective effect on *E. coli* 25,922 cells challenged with PANI or P3ABA suspension in rich media. $H_2O_2$ is membrane permeable and will equilibrise across the cell membrane meaning that exogenous $H_2O_2$ levels in growth media will reflect the amount of endogenous $H_2O_2$ (that has not been scavenged and is

able to cause damage) (*Ravindra Kumar & Imlay, 2013*). Catalase, which scavenges $H_2O_2$ generating water and oxygen, was added to *E. coli* 25922 cells exposed to PANI or P3ABA suspension in LB broth (*Bhaumik et al., 1995*). $H_2O_2$ was used as a positive control. Catalase is membrane impermeable and therefore can only detoxify exogenous $H_2O_2$ (*Bhaumik et al., 1995*). Catalase detoxifying $H_2O_2$ in the growth media would cause a decrease in intracellular $H_2O_2$ levels as it equilibrises across the cell membrane. Therefore, exogenous catalase can protect against intracellular $H_2O_2$-mediated damage. A protective effect by catalase (reflected in the viable cell count) is indicative of excess $H_2O_2$ production in the exposed cells while no protective effect suggests that induction of $H_2O_2$ based oxidative stress is not occurring.

It was hypothesised that *E. coli* cells would be less sensitive to PANI treatment in the presence of catalase. $H_2O_2$ exposed cells dropped below the limit of detection while $H_2O_2$ exposed cells incubated with catalase were present at a similar level to the untreated controls (Fig. 5A). PANI treated cells showed a similar trend to the $H_2O_2$ treated cells. The treated cells incubated with catalase were present at higher numbers (median cell count: $8.5 \times 10^3$ CFU/ml) compared to those incubated without catalase (median cell count: $1 \times 10^2$ CFU/ml), suggesting that catalase protected against PANI action by reducing exogenous (and therefore endogenous) $H_2O_2$ levels (Fig. 5A). These results support PANI antimicrobial action involving production of $H_2O_2$ and induction of oxidative stress.

It was hypothesised that exogenous catalase would provide some protection to *E. coli* against P3ABA antimicrobial action in rich media. *E. coli* cells treated with $H_2O_2$ were below the limit of detection at the time points used while the presence of exogenous catalase prevented the decrease in viable cell counts (Fig. 5B). In contrast to the trend seen for PANI, P3ABA treated cells were knocked down independent of the presence of catalase. Similar viable cell counts were obtained for P3ABA alone (median cell count: $1.51 \times 10^4$ CFU/ml) and P3ABA with catalase ($9.49 \times 10^3$ CFU/ml) (Fig. 5B). These results indicate that production of excess $H_2O_2$ is not occurring in P3ABA treated cells. Therefore, it is apparent that induction of $H_2O_2$-based oxidative stress does not have a major role in P3ABA antimicrobial action.

## Antimicrobial mechanism of PANI involves ROS production

The sensitivity profile of PANI to Keio collection deletion mutants reinforced the hypothesis that antimicrobial action involves production of ROS. In rich media, the supersensitivity of *E. coli* $\Delta katG$, which is unable to scavenge exogenous $H_2O_2$, and the decreased sensitivity of *E. coli* $\Delta ahpC$, which is postulated to be preadapted to oxidative stress, supported this hypothesis (Table 1). There was also evidence to suggest perturbation of iron homeostasis in PANI treated cells in iron rich media as demonstrated by the reduced sensitivity of *E. coli* $\Delta tonB$, which has reduced internal iron levels (Table 1). Iron homeostasis is intimately related to oxidative stress as free iron can propagate $H_2O_2$ stress by participating in Fenton reaction producing highly damaging hydroxyl radicals (*Imlay, 2013*). The increased susceptibility of *E. coli* $\Delta iscS$ to PANI treatment (Table 1) may be due to an inability to modify DNA to protect against oxidation, suggesting involvement of oxidative stress

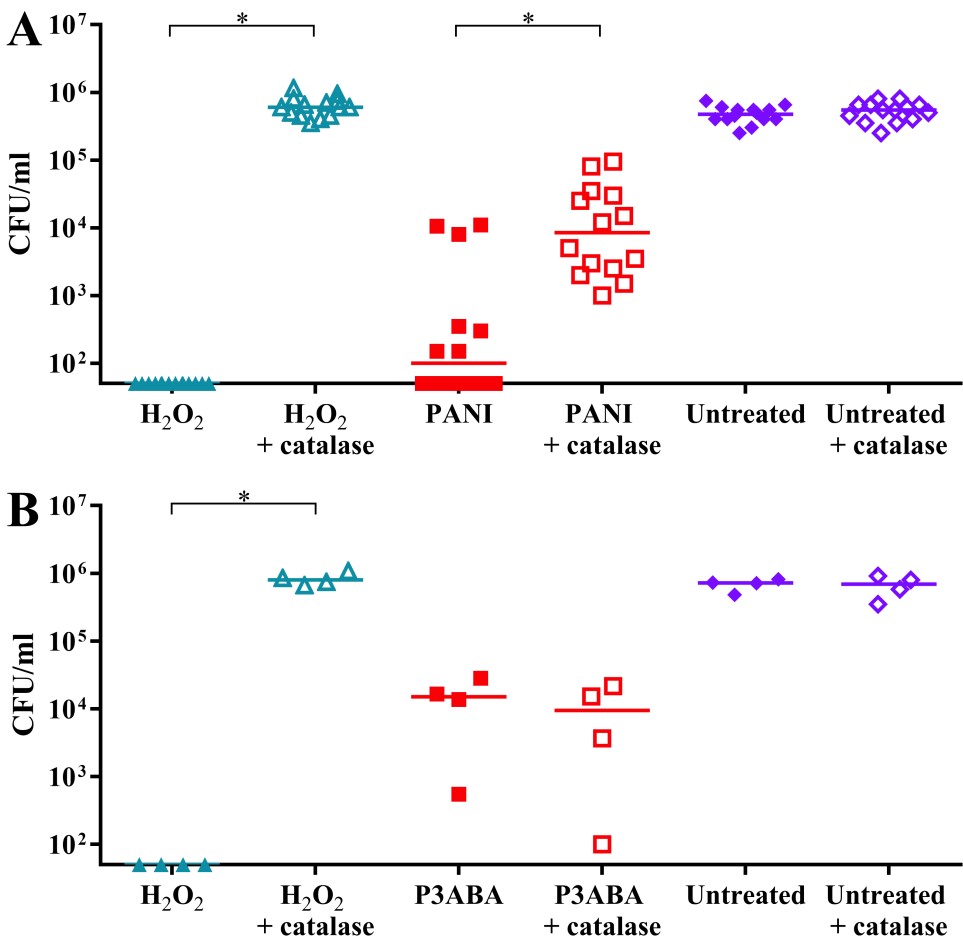

**Figure 5** **The effect of exogenous catalase on the sensitivity of *E. coli* 25922 to the antimicrobial action of PANI and P3ABA.** Cell viability assay of ~$10^6$ CFU/ml *E. coli* 25922 challenged with 2.5 mM $H_2O_2$, 2% PANI suspension or 0.5% P3ABA suspension in LB broth. The treatment times for PANI and P3ABA were 2 h and 0.5 h, respectively, with $H_2O_2$ treated and untreated cells enumerated at the same time. Each data point represents a biological replicate with a line plotted at the median. Statistical significance is represented by * (repeated measures ANOVA, *P* value: less than 0.05).

(*Xie et al., 2012*; *Imlay, 2013*). There was also evidence of periplasmic stress as indicated by the supersensitivity of *E. coli* Δ*spy* to PANI suspension (Table 1).

The role of induction of oxidative stress in the mode of action of PANI was further investigated in *E. coli* 25922 by examining the activity of the antimicrobial agent in aerobic and anaerobic conditions. PANI was able to exert greater lethal activity when oxygen was present compared to anaerobic conditions in both rich and minimal media (Figs. 3A–3C). Endogenous ROS are generated at increased rates when the concentration of oxygen is high and cannot be produced when oxygen is absent (*Seaver & Imlay, 2004*; *Ravindra Kumar & Imlay, 2013*). Therefore, it was concluded that the greater activity of PANI in aerobic conditions reflects an oxidative stress mechanism involving increased production of ROS in treated bacterial cells, which may mediate bactericidal action (*Dwyer et al., 2014*).
Furthermore, aerobic bactericidal activity of PANI against *E. coli* cells in minimal media with glycerol as the carbon source was greater than that in minimal media with succinate (Figs. 3B–3C). The difference in aerobic susceptibility to PANI could be indicative of the high concentration of succinate ($\sim$15 mM) reducing endogenous $H_2O_2$ formation by preventing fumarate reductase autoxidation and thus protecting against the putative PANI antimicrobial action (*Messner & Imlay, 2002a*). The antimicrobial activity of PANI is more affected by the presence of oxygen compared to P3ABA, which supports the hypothesis that PANI mediates induction of oxidative stress (Fig. 4).

Additional support for the hypothesis that PANI exerts bactericidal activity through facilitating $H_2O_2$ production was derived from determining the protective effect of exogenous catalase. Membrane impermeable catalase can detoxify endogenously produced $H_2O_2$ as it equilibrises across the bacterial cell membrane (*Ravindra Kumar & Imlay, 2013*). Therefore, exogenous catalase can protect against intracellular $H_2O_2$-mediated damage. Catalase protected *E. coli* cells against PANI bactericidal action (Fig. 5A), which is indicative of excess $H_2O_2$ production in the exposed cells.

The mechanistic basis of PANI antimicrobial activity is hypothesised to be electrical conductivity, which can mediate contact with the negatively charged bacterial cell surface through electrostatic adherence and may be responsible for increased production of $H_2O_2$ (*Gizdavic-Nikolaidis et al., 2011*). Low concentrations of colloidal PANI have been associated with biocompatibility and ROS scavenging abilities while higher concentrations have been associated with antimicrobial activity, which demonstrates how the degree of electrical conductivity can influence organism viability (*Kucekova et al., 2014*). PANI may perturb the flow of electrons through the ETC (potentially mediated through electron acceptor capabilities), which would result in accumulation of reducing equivalents leading to accelerated side reactions with oxygen, generating ROS (*Akhova & Tkachenko, 2014*). $H_2O_2$ is formed inside aerobically growing *E. coli* is 10–15 $\mu$M per second but is effectively scavenged by alkyl hydroperoxide reductase and catalase (*Imlay, 2013*). If $H_2O_2$ levels increase past the threshold that can be scavenged, then potentially lethal damage will occur.

Many metabolic and respiratory enzymes are inactivated following oxidation by $H_2O_2$ within minutes (*Imlay, 2014*). Susceptible enzymes have solvent exposed iron atoms and include those containing $[4Fe-4S]^{2+}$ clusters (dehydratase enzymes, such as fumarase A and fumarase B from the TCA cycle) and mononuclear iron enzymes (such as 3-deoxy-D-arabinoheptulosonate 7-phosphate synthase involved in aromatic compound biosynthesis) (*Jang & Imlay, 2007*; *Sobota, Gu & Imlay, 2014*). Oxidation of the iron atom promotes its dissociation from the enzyme, causing loss of function, and an increase in free iron levels (*Djaman, Outten & Imlay, 2004*). Increased free iron released from oxidised enzymes can participate in Fenton reaction producing hydroxyl radicals (*Linley et al., 2012*). The decreased sensitivity of *E. coli* Δ*tonB* (Table 1) to PANI suspension supports the involvement of free iron in antimicrobial action.

Fur acts to control intracellular free iron levels by suppressing iron uptake and promoting iron utilisation. Iron bound Fur can be oxidised by $H_2O_2$ inactivating the repressor (*Varghese et al., 2007*). OxyR increases the expression of the *fur* gene in response to this; however, if the $H_2O_2$ levels are excessively high, most Fur would be inactivated

leading to derepression of iron acquisition systems and the deleterious import of more iron (*Varghese et al., 2007*). Inactivation of Fur would release RyhB from Fur-mediated repression preventing synthesis of proteins involved in iron utilisation and storage, including *sdhCDAB* (*Massé & Gottesman, 2002*; *Massé, Escorcia & Gottesman, 2003*; *Massé & Arguin, 2005*). Thus, $H_2O_2$ can cause free iron levels to increase due to targeting of oxidation susceptible iron containing enzymes and deregulation of iron homeostasis involving uncontrolled iron import and a decreased capacity for incorporation of iron into proteins. Upregulation of *ryhB* expression was found to occur in response to fPANI treatment, which supports de-repression of *ryhB* expression due to oxidative inactivation of Fur (*Gizdavic-Nikolaidis et al., 2011*). Further increase in intracellular free iron levels would result in the production of more hydroxyl radicals and exacerbate the oxidative stress (*Varghese et al., 2007*).

Hydroxyl radicals are potent oxidants that can damage biomolecules, including DNA, proteins, and lipids (*Linley et al., 2012*). It is postulated that the lethality of $H_2O_2$ is due to ROS-mediated DNA double-strand breaks, particularly due to oxidation of guanine and thymine nucleotides (*Foti et al., 2012*; *Dwyer, Collins & Walker, 2015*). DNA repair systems function to detect and repair damage; however, if these systems are overwhelmed then damage is not repaired (*Imlay, 2013*). Increased hydroxyl radicals derived from elevated free iron levels would result in increased rates of DNA damage (*Linley et al., 2012*). Thus, oxidative stress propagated by free iron can cause lethal DNA damage.

The involvement of DNA in the antimicrobial mechanism of PANI was investigated by challenging DNA repair mutant, *E. coli* Δ*recA*. The decreased sensitivity of this mutant strain to PANI may reflect mode-2 $H_2O_2$ mediated killing as opposed to mode-1 killing, which involves DNA damage. Mode-2 killing occurs at higher $H_2O_2$ concentrations (more than 10 mM), involves Fenton reaction and hydroxyl radicals, and reflects general damage to cellular biomolecules (*Imlay & Linn, 1987*; *Martinez & Kolter, 1997*; *Linley et al., 2012*). Mode-2 killing does not require active growth, which is supported by the efficacious activity of PANI in minimal media against *E. coli* (Figs. 3B–3C). This type of $H_2O_2$ killing is reduced in anoxic conditions, which is consistent with the reduction in activity observed for PANI in anaerobic conditions (Fig. 4) (*Brandi et al., 1987*; *Linley et al., 2012*). Therefore, PANI may mediate production of a high concentration of $H_2O_2$ resulting in mode-2 killing characterised by damage to proteins and/or the cell membrane. Oxidation of amino acids in proteins can cause a range of outcomes, from total degradation of the protein backbone to minor side-chain modification of individual residues (*Dean et al., 1997*). Loss of essential protein function may cause bacterial cell death.

There is a range of evidence to suggest that PANI mode of action includes production of $H_2O_2$ and the consequences of this; however, it is likely that there are additional mechanisms involved. The lack of significant difference between the tested *E. coli* deletion mutants and the parent strain challenged with PANI in minimal media suggests that other unidentified mechanisms are involved in the antimicrobial action. Also, PANI activity was reduced in anaerobic conditions compared to aerobic conditions (Figs. 3A–3C) but bactericidal activity was not completely abolished. Thus, in anaerobic conditions there

must be other unidentified mechanisms occurring to cause cell death, the impact of which would likely vary relative to environmental conditions (*Dwyer et al., 2014*).

## P3ABA acts as an uncoupler

The sensitivity profile of *E. coli* deletion mutants challenged with P3ABA suspension inferred that the antimicrobial mechanism involves perturbation of metabolism and iron homeostasis, periplasmic stress, and acid stress. It should be noted that there may be additional antimicrobial mechanisms occurring that may have been missed due to testing of only a selection of deletion mutants. There was evidence to suggest (Table 1) oxidative damage (*E. coli* Δ*grxA*) to metabolic and respiratory enzymes (*E. coli* Δ*sdhB* and *E. coli* Δ*iscS*) occurs leading to perturbation of iron homeostasis (*E. coli* Δ*fur* and *E. coli* Δ*tonB*) (*Touati et al., 1995*; *Schwartz et al., 2000*; *Starke, Chock & Mieyal, 2003*; *Vlamis-Gardikas, 2008*). The increased sensitivity of *E. coli* Δ*asr* to P3ABA (Table 1) indicates that acid stress occurs during P3ABA antimicrobial action (*Suziedeliené et al., 1999*). The decreased sensitivity of as *E. coli* Δ*spy* in LB broth to P3ABA action (Table 1) probably reflected upregulation of an additional extracytoplasmic stress response (*Raivio & Silhavy, 2001*), possibly from acid-induced protein misfolding. ATP synthase was identified as a potential target of P3ABA action as *E. coli* Δ*atpE* in rich media had reduced sensitivity to P3ABA treatment compared to the parent strain (Table 1). Bedaquiline, a novel antitubercular drug, is known to target ATP synthase, acting as an uncoupler (*Sun et al., 2012*; *Koul et al., 2014*; *Hards et al., 2015*). *M. tuberculosis* has been demonstrated to be sensitive to the antimicrobial action of P3ABA, which supports the ascertain that P3ABA acts as an uncoupler (*Robertson et al., 2016*).

The involvement of perturbation of iron homeostasis in the antimicrobial mechanism of P3ABA was further investigated by using EPR spectroscopy to measure internal free iron levels in sublethally treated cells. An increase in intracellular free iron was detected in P3ABA treated cells in LB broth, similar to, but in a greater magnitude than $H_2O_2$ treatment (Fig. 2). Hydroxyl radicals are formed when unincorporated iron reacts with $H_2O_2$ during Fenton reaction (*Imlay, 2013*). The perturbation of iron homeostasis could promote oxidative stress, particularly the formation of hydroxyl radicals, which could explain the increased sensitivity of *E. coli* Δ*grxA* (Table 1). Furthermore, the lack of significant difference in sensitivity between the parent strain and *E. coli* Δ*tonB* in LB broth suggests that the increase in intracellular free iron that occurs in P3ABA treated cells is derived from internal, rather than external, sources (*Massé & Arguin, 2005*; *Dwyer et al., 2007*).

The activity of P3ABA in aerobic and anaerobic conditions against *E. coli* 25922 was examined to determine the involvement of ROS in the mechanism of P3ABA. *E. coli* 25922 was challenged with P3ABA in both rich and minimal media, the latter with glycolytic and non-glycolytic carbon sources, allowing for inferences to be made relating to the effect of the energy state of the cell on the susceptibility to P3ABA action. Involvement of oxidative stress, indicated by greater activity in aerobic conditions, was only associated with *E. coli* challenged in rich media and not in minimal media (Figs. 3D–3F). Production of ROS has been shown to be a downstream effect of futile cycling caused by uncoupling activity (*Adolfsen & Brynildsen, 2015*; *Hards et al., 2015*). P3ABA was more active against

*E. coli* in rich media compared to minimal media in aerobic conditions, which is consistent with increased susceptibility to futile cycling and bactericidal action associated with high respiration rates (*Lobritz et al., 2015*).

There was no difference in the activity of P3ABA when the *E. coli* cells were challenged in minimal media with a glycolytic carbon source (glycerol) compared to a non-glycolytic carbon source (succinate), from which ATP can only be made by oxidative phosphorylation, signifying that the ability to synthesise ATP by alternate methods does not prevent P3ABA bactericidal action (*Jensen et al., 1995*; *Hards et al., 2015*). A similar trend has been observed for *M. smegmatis* treated with bedaquiline, an inhibitor of ATP synthase, supporting the hypothesis of P3ABA acting as an uncoupler (*Hards et al., 2015*).

P3ABA is postulated to target, and probably damage, ATP synthase. This targeting is likely to be due to the functionalisation of P3ABA as there was no evidence to suggest that PANI targets ATP synthase. Damage to this biomolecule may permit $H^+$ ions to move from the periplasmic space into the cytoplasm in an uncontrolled manner that is uncoupled from ATP synthesis (*Hards et al., 2015*). The cell would continue to respire and the ETC would continue to pump $H^+$ into the periplasm as electrons are transported along the chain, resulting in a futile proton cycle (*Hards et al., 2015*). In response to the futile cycling, respiration and oxygen consumption would increase potentially resulting in elevated production of ROS (*Adolfsen & Brynildsen, 2015*; *Belenky et al., 2015*).The effects of increased ROS production in a bacterial cell include dysregulation of iron homoeostasis. The production of ROS is likely a downstream effect, which would explain why it was not suggested by deletion mutant screening (Table 1) or by examining the potential protective effect of exogenous catalase (Fig. 5B).

P3ABA mediated damaged to ATP synthase may cause acid stress as a downstream effect as indicated by the susceptibility of *E. coli* Δ*asr* to P3ABA (Table 1). ATP synthase and other respiratory machinery components are implicated in the regulation of internal pH, which implies that damage to ATP synthase and perturbation of respiration would sensitise the cells to acid stress (*Sun et al., 2012*).

The exact events that directly lead to P3ABA-mediated cell death are not known. The uncontrolled leakage of protons may cause dissipation of proton motive force (PMF; the sum of the transmembrane pH gradient and the electrical potential), which is lethal to all living cells (*Rao et al., 2008*; *Hards et al., 2015*). Supportive of this is the rapid loss of membrane potential in *E. coli* that occurs following P3ABA exposure (*Robertson, 2012*). Collapse of PMF in *E. coli* and *S. aureus* has been shown to trigger the process of autolysis (*Brunskill & Bayles, 1996*; *Lamsa et al., 2012*).

A proposed model for this, as described for *Bacillus subtilis*, involves repression of murein hydrolase activity in cells with an intact PMF mediated by the localised reduction in pH within the cell wall (*Bayles, 2003*; *Rice & Bayles, 2008*; *Tanouchi et al., 2013*). Following dissipation of membrane potential, the cell wall pH increases resulting in derepression of the murein hydrolases (*Lamsa et al., 2012*). Deregulated murein hydrolases degrade peptidoglycan in the bacterial cell wall, disrupting cell wall integrity and resulting in cell lysis (*Lamsa et al., 2012*). It remains to be investigated whether autolysis occurs in P3ABA treated cells. Bactericidal action of bedaquiline, which seems to act in a similar manner to

P3ABA, does not involve dissipation of PMF, attributed to the coupled exchange of other cations maintaining the electrical potential (*Hards et al., 2015*). Elucidation of the lethal events in bedaquiline action is still ongoing and may inform potential events in P3ABA bactericidal action (*Cook et al., 2014b*).

## CONCLUSION

PANI in rich media is postulated to exert antimicrobial activity through increasing $H_2O_2$ levels leading to oxidative stress characterised by perturbation of iron homeostasis, Fenton reaction, and damage by hydroxyl radicals. P3ABA in minimal media is proposed to reduce bacterial cell viability by targeting ATP synthase causing uncontrolled proton leak into the cytoplasm. The resulting futile cycling may cause dissipation of PMF as well as acid stress as a downstream effect. While it is apparent that the antimicrobial mechanism of P3ABA involves targeting of ATP synthase, the events that occur following this are not fully characterised and require further investigation.

## ACKNOWLEDGEMENTS

The authors thank Sudip Ray, Adeline Le Cocq, Chris Wilcox and Walt Wheelwright for purified PANI and P3ABA.

### Funding

The authors received research funding from the New Zealand Ministry of Business, Innovation and Employment (MBIE) for research programmes UOAX0812 and UOAX1410, and the University of Auckland's Vice Chancellors Strategic Development Fund, grant number 23563. The funders had no role in study design, data collection and analysis, decision to publish, or preparation of the manuscript.

### Grant Disclosures

The following grant information was disclosed by the authors:
New Zealand Ministry of Business, Innovation and Employment (MBIE): UOAX0812 and UOAX1410.
University of Auckland's Vice Chancellors Strategic Development Fund: 23563.

### Competing Interests

The authors declare there are no competing interests.

### Author Contributions

- Julia Robertson conceived and designed the experiments, performed the experiments, analyzed the data, prepared figures and/or tables, authored or reviewed drafts of the paper, approved the final draft.
- Marija Gizdavic-Nikolaidis conceived and designed the experiments, analyzed the data, contributed reagents/materials/analysis tools, authored or reviewed drafts of the paper, approved the final draft.

- Michel K. Nieuwoudt performed the experiments, analyzed the data.
- Simon Swift conceived and designed the experiments, analyzed the data, authored or reviewed drafts of the paper, approved the final draft.

## Data Availability

The raw data are provided in Data S1.

## Supplemental Information

Supplemental information for this article can be found online at http://dx.doi.org/10.7717/peerj.5135#supplemental-information.

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
