# Peer review of "The antimicrobial action of polyaniline involves production of oxidative stress while functionalisation of polyaniline introduces additional mechanisms"

_PeerJ, doi:10.7717/peerj.5135_

## Round 0.1 · original submission · Major Revisions

The Reviewers have made suggestions and comments that, I believe, will help to improve the manuscript. Please pay special attention to the statistical analysis and presentation of the results and the data presented in Table 1 that both Reviewer 2 and 3 have commented on. Please make appropriate changes and submit your revised manuscript.

Reviewer 1 ·

Basic reporting

The manuscript is well prepared. Literature are actual and relevant. Figures have adequate quality.

Experimental design

Experimental design as well as used methods are relevant.

Validity of the findings

The novelty of the results are high. Conclusions are based on the results.

Additional comments

Manuscript „The antimicrobial action of polyaniline involves production of oxidative stress while functionalization of polyaniline introduces additional mechanisms“ focused on highly discussed topic regarding the biological properties of polyaniline. The study is well designed and all used methods are relevant. The manuscript provide new results which can significantly contribute to our understanding of antibacterial effect of PANI. The conclusions are based on gained results. I have only one minor suggestion: the results show the PANI activity is connected to the production of ROS. It could be interesting to discuss this more even in context to the previously found impact of PANI on eukaryotic cells (e.g. http://dx.doi.org/10.1016/j.colsurfb.2014.01.027). I have no substantial concerns about this manuscript and therefore I do recommend it for publication in its present form.

·

Basic reporting

The manuscript describes the antibacterial activity of polyaniline (PANI) and and a homopolymer of poly(3-aminobenzoic acid). Activity on single gene deletion mutants was used to demonstrate that these two molecules have different effects on E. coli cells. The introduction provides sufficient back ground information and the use of English is clear and professional. There are a few points that need to be addressed. One of the major issues is that there is a lot of discussion included in the results, for example the discussion of bedaquiline. This makes the results section difficult to follow. If two separate sections are going to be provided they need to be separate. Additional point that need clarification are below
Line 325. It is unclear what the statement that “the E. coli deletion mutants were more affected” means. Was the phenotype more pronounced? Were more strains affected?
Line 330-365. The detailed discussion of each strain is unnecessary. This was not done for the strains with altered sensitivity in the other experiments and should be summarized similarly.
Line 367. The results of the mutant screening indicate that PANI likely induces an increase in free iron but the results from P3ABA in minimal media indicate that the cells are likely undergoing iron starvation.
Line 402. Product needs to be changed to production
Line 427. It would support the claims if the values were included in the text. This applies throughout the paper.
Line 429-434. Is this the same statement twice or are these supposed to provide different comparisons. Either way the meaning of these statements needs to be clarified.

The results section describing the activity of PANI and P3ABA in aerobic vs anaerobic conditions is very hard to follow. This section would benefit from a more focused approach to drive home the point the authors are trying to highlight from this data. Alternatively this could be separated into multiple sub sections to make the conclusions more clear.

Experimental design

The methods are for the most part well described and appropriate for each experiment. However I do have some concerns about the rationale for certain experiments.
Line 122. It would be helpful to include a brief statement on the rationale used to select the deletion mutants. Why not include some with deletions in other cellular functions as controls?
Line 292. Again how was this subset selected?
Table 1. The authors should provide the data used to determine the sensitivity of each strain as opposed to merely stating less or more sensitive. This needs to include statistical analysis.
Why were the EPR experiments only done in LB?
There are a lot of claims made about the induction of ROS using indirect evidence. Detecting ROS in cells is a relatively simple experiment to do using probes such as DHR123 or DCFDA. Is there a reason these experiments have not been included?
Figure 4. The data is presented in a very confusing way. It would be helpful to change the y axis to log2 of the fold change in MBC or something similar

Validity of the findings

The findings described in this manuscript suffer from a lack of focus in their description. In the introduction a simple question was posed on the difference in the mechanism of two antimicrobials. There is a lot of unnecessary information in the results section which makes it difficult to follow. There are also some comparisons that take away from the initial question posed. A reorganisation of the results and discussion may help with the inclusion of more headings that highlight the key finding similar to the section “P3ABA treatment causes an increase in free iron in E. coli”. Some specific concerns are listed below.

Line 391. The EPR data that shows the increase in free iron in P3ABA treated cells contradicts the statement in line 314-315 where the authors found no change in sensitivity to P3ABA in strains with deletions in genes related to iron homeostasis. This needs to be explained

The authors need to acknowledge that only screening a small subset of deletion strains biases their results. It is possible that there are other mechanisms at play that have been missed due to the strains selected. This is particularly evident in the lack of altered sensitivity of any strain to PANI in minimal media. There must be another mechanism at play that could also be contributing in rich media. Similarly there are only two strains with altered sensitivity to P3ABA in LB.

The conclusion statement over simplifies the findings of the paper. The mechanisms described for PANI is the one identified in LB whereas the one for P3ABA is the one for minimal media.

Reviewer 3 ·

Basic reporting

Roberson and colleagues perfomed studies to determine the mechanism of action of the novel antimicrobial agents polyaniline (PANI) and functionalised polyanilines (fPANI). Their work shows that PANI antimicrobial activity is due to hydrogen peroxide production and consequent oxidative damage, while P3ABA (fPANI) targets ATP synthase, disrupting respiration and metabolic processes.

I commend the authors for their extensive data set (including the raw data file). The manuscript is well written, with detailed methods and a good state of the art introduction.
If there is a weakness, it is in the lack of independent replicates for some experimental procedures, which should be improved upon before Acceptance.

Experimental design

Materials and Methods section was well written and detailed.

1. In general, some of the methodology was not performed in 3 independent experiments, which weakens the work. For example, the EPR iron determination was assessed only once (n =1) with 2 biological replicates.
2. A statistical analysis paragraph compiling all analysis made for the entire methodology would help the reader following the materials and methods section.

Validity of the findings

1. The initial screening of the mutant’s Keio library was very helpful to pinpoint the most plausible mechanisms behind the PANI and P3ABA mode of action. However, I had trouble understanding the results in Table 1. It was very confusing because there are a lot of bank spaces (which I supposed meant there was no difference compared with the parental strain). The data set could have a better visual aid by using a column plot for example.
Have the authors tested whether each mutant strain has growth defects in both media, without any treatment?

2. Free iron determination by EPR is lacking independent replication of the data (Figure 2), with only one experiment performed (n=1). This strongly makes me question this result. I would encourage repeating again to get (at least) an n = 2 independent replicates.

3. Again, in Figure 3, no information about n, SD (standard deviation) and statistic analysis made (I presume the Mann-Whitney is the non-parametric Mann-Whitney U-test but this should be in the statistic analysis in the materials and methods).

In general, the findings are well supported by the data presented in figures and in the supplemental materials.

Additional comments

In general, figure captions were lacking important information regarding the data set (e.g. statistical analysis, n number, standard deviation, concentration of treatments).

line 470 – bedaquiline is mentioned for the first time and the authors only explain what it is later in line 483.

line 774 – The overall conclusion that P3ABA may cause PMF collapse, when no experimental data investigating membrane damage and deregulation of murein hydrolases, is too bold of claim.

The authors’ recently published work (Robertson J, Gizdavic-Nikolaidis M, Swift S. Investigation of Polyaniline and a Functionalised Derivative as Antimicrobial Additives to Create Contamination Resistant Surfaces. Materials. 2018;11(3):436. doi:10.3390/ma11030436) should be cited in this manuscript since it has relevant data on another tested bacteria species (Staphylococcus aureus) which further demonstrates the antimicrobial effectiveness of these compounds against not only E. coli but also S. aureus, in addition to tests for surface application.

---

## Round 0.2 · accepted · Accept

Thank you for carefully considering each of the Reviewers' comments, preparing in-depth responses to their questions and concerns, and editing your manuscript accordingly. It has been a pleasure working with you!

#